# Research on Energy Harvesting Mechanism and Low Power Technology in Wireless Sensor Networks

**DOI:** 10.3390/s24010047

**Published:** 2023-12-21

**Authors:** Weimin Chen, Feng Tang, Fang Cui, Chen Chen

**Affiliations:** 1School of Information and Electronic Engineering, Hunan City University, Yiyang 413000, China; chenweimin@hncu.edu.cn; 2School of Software Engineering, South China University of Technology, Guangzhou 510006, China; cc030328@163.com; 3School of Computer Science and Engineering, Central South University, Changsha 410083, China; cf2695@163.com

**Keywords:** wireless sensor networks, energy harvesting, neighbor discovery, data fusion

## Abstract

Wireless sensor networks (WSNs) are widely used in various fields such as military, industrial, and transportation for real-time monitoring, sensing, and data collection of different environments or objects. However, the development of WSNs is hindered by several limitations, including energy, storage space, computing power, and data transmission rate. Among these, the availability of power energy plays a crucial role as it directly determines the lifespan of WSN. To extend the life cycle of WSN, two key approaches are power supply improvement and energy conservation. Therefore, we propose an energy harvesting system and a low-energy-consumption mechanism for WSNs. Firstly, we delved into the energy harvesting technology of WSNs, explored the utilization of solar energy and mechanical vibration energy to ensure a continuous and dependable power supply to the sensor nodes, and analyzed the voltage output characteristics of bistable piezoelectric cantilever. Secondly, we proposed a neighbor discovery mechanism that utilizes a separation beacon, is based on reply to ACK, and can facilitate the identification of neighboring nodes. This mechanism operates at a certain duty cycle ratio, significantly reduces idle listening time and results in substantial energy savings. In comparison to the Disco and U-connect protocols, our proposed mechanism achieved a remarkable reduction of 66.67% and 75% in the worst discovery delay, respectively. Furthermore, we introduced a data fusion mechanism based on integer wavelet transform. This mechanism effectively eliminates data redundancy caused by spatiotemporal correlation, resulting in a data compression rate of 5.42. Additionally, it significantly reduces energy consumption associated with data transmission by the nodes.

## 1. Introduction

Wireless sensor networks (WSNs) are a soaring frontier research hotspot in multiple subject areas [1]. A WSN can collaboratively monitor, sense, and collect the various information of environments or monitored objects in real time through various integrated microsensors, and transmits the perceived information to the user terminal by multihop relay through a random self-organized wireless communication network. WSNs can realize the idea of “ubiquitous computing”, and have become an integral part of the C4ISRT system (command, control, communication, computing, intelligence, surveillance, reconnaissance, and targeting). Today, it is widely used in various areas of IoT/AI, such as healthcare [2], energy management [3], smart transportation [4,5,6], thermal comfort [7], and energy load identification [8].

The nodes of a WSN are usually a miniature embedded system. Its advantages are small size, low cost, and easy to deploy, while the disadvantages are that its energy, storage space, computing power, and bandwidth are greatly limited. Most WSNs are deployed in sparsely populated or human-inaccessible areas, and it is difficult to replace the batteries for the nodes. Therefore, the limited power energy is one of the most important constraints in the design of the whole WSN, which directly determines the life cycle and application life of the network. To extend the life cycle of WSN, there are two ways: power supply and energy saving. For this reason, many researchers at home and abroad have carried out a lot of work, such as investigating self-power supply technology for harvesting energy from nature [9], wireless power technology [10], and the node energy consumption problem [11]. However, their proposed schemes lack comprehensiveness and exhibit poor performance. Additionally, certain schemes may be too computationally demanding to be implemented on sensor nodes that have limited resources. To prolong the life cycle of WSNs, we focused on three key areas for improvement: (1) Energy harvesting technology. We analyzed the current mechanisms and methods of energy harvesting technology, explored the utilization of solar energy and mechanical vibration energy to ensure a continuous and dependable power supply to the sensor nodes, and examined the voltage output characteristics of a bistable piezoelectric cantilever in order to gain further insights. (2) Low-duty-cycle working mode. We proposed a neighbor discovery mechanism that utilizes a separation beacon, based on reply to ACK, to facilitate the identification of neighboring nodes. This mechanism operates at a certain duty cycle ratio, significantly reducing idle listening time and resulting in substantial energy savings. (3) Data fusion technology. We introduced a data fusion mechanism founded on integer wavelet transform. This mechanism effectively eliminates data redundancy caused by spatiotemporal correlation and can reduce energy consumption associated with data transmission by the nodes.

## 2. Related Work

The energy management of nodes can increase the service cycle and reduce the cost of a WSN. It involves two problems, namely, energy supply and energy consumption. The research progress at home and abroad is described below.

### 2.1. Energy Harvesting Technology

There are two main energy supply methods for the network nodes: wireless charging technology and energy harvesting technology. However, the wireless charging technology has limitations in terms of short charging distances and is not suitable for powering WSNs deployed in remote areas [12,13]. Fortunately, the energy harvesting technology provides an ideal solution for meeting the energy demands of WSNs [14]. Currently, there exist numerous well-developed energy harvesting technologies. Of all these technologies, solar energy harvesting stands out as the earliest and most advanced method. The solar harvester has the ability to generate milliwatts of energy per square centimeter, making it suitable for powering wireless sensor nodes. However, its performance is affected by environmental factors such as darkness and precipitation. On the other hand, vibration energy is readily abundant in the natural environment and has a high energy density, making it a perfect candidate for environmental energy harvesting technology [15]. Among the different types of vibration energy harvesting devices, piezoelectric devices have garnered considerable interest in recent years [16,17]. This is primarily due to their exceptional attributes, such as high energy conversion efficiency, utilization of lightweight materials, simple yet robust structure, and seamless integration. Currently, the utilization of piezoelectric devices is predominantly based on the implementation of cantilever, cymbal, and cylindrical structures for energy harvesting. Among these structures, the cantilever structure has garnered considerable interest in both domestic and international research circles due to its straightforward implementation and resonance capabilities in low-frequency natural environments [18,19].

### 2.2. Low-Duty-Cycle Working Mode

In this mode, the radio frequency (RF) of nodes will enter the active state that can send and receive data, and be closed and turned into the dormant state at other time to reduce energy consumption. At this point, the mutual discovery between adjacent nodes becomes the main problem. There are classical neighbor discovery protocols at home and abroad, including Disco, U-Connect, Hello, and Nihao [20]. They focus on optimizing the scheduling period when the node works to find neighboring nodes to each other as quickly as possible with as few active time slots as possible. Y Zhang et al. [21] demonstrated that there is less likely to be a conflict at the beginning and end of the time slot, and proposed that the probability of conflict can be reduced by dynamically changing the time slot length and making random retreat. K Bian et al. [22] proposed that beacon conflict can be reduced by controlling the length of slot time and the number of beacons. S Jin et al. [23] proposed a mixed mode of send–receive–send and send–receive, and explored the relationship between the discovery performance and slot size on a specific hardware platform. Y Qiu et al. [24] proposed an interactive mechanism by separating the sending and reception of a beacon. In this mechanism, nodes only send the beacon in the transmission time slot and receive the beacon in the listening time slot, which can reduce the probability of beacon conflict.

### 2.3. Data Fusion Mechanism

Saeedi et al. [25] demonstrated, using data fusion technology, that the larger the network or the higher the number of source nodes, the more significant the effect of saving energy is. There are various classifications of data fusion from different perspectives. Based on the spanning tree, Wensheng Zhang et al. [26] proposed the DCTC (dynamic convey tree-based collaboration) algorithm. The convergence nodes perform data fusion on the data of their subgenerating tree nodes. Hong Luo et al. [27] proposed the MFST (minimum fusion Steiner tree) algorithm for energy-efficient data collection in data fusion mode in WSNs. Based on the spatiotemporal correlation, the TiNA model (temporal coherency aware in network aggregation) was proposed [28]. Its basic idea is that the node sends the data only when the difference between the currently collected data and the last collected data is greater than the tolerance limit specified by a certain user. The spatial fusion model [29] measures the correlation by the distance between nodes. The spatiotemporal fusion model [30] is the development trend of data fusion research to eliminate spatiotemporal correlation. The LEACH protocol (low-energy adaptive clustering hierarchy) [31] is a typical WSN routing protocol which forms a hierarchical structure-based routing mechanism.

As presented above, the energy harvesting technology and energy saving mechanisms (including low-duty cycle and data fusion) are the two main means to extend the service life of wireless sensor networks. However, in previous studies, they often focused on only one technology and did not comprehensively consider the problem of extended service life span of wireless sensor networks. Based on this case, we comprehensively used technologies such as energy harvesting, neighbor discovery and data fusion to analyze the energy supply and energy consumption of nodes, and hoped to propose a new approach to achieve a reliable and efficient way to secure energy in WSNs.

## 3. Methods

To enhance the energy efficiency and longevity of the network, we implemented two strategies for the WSN, as illustrated in Figure 1. Firstly, we employed energy harvesting technology to convert renewable sources like solar and wind energy into electrical energy, which effectively supplements the power supply of the nodes. Secondly, we utilized neighbor discovery technology to decrease the operational time of the nodes, along with data fusion technology to reduce the volume of data transmission. These measures were aimed at minimizing energy consumption and prolonging the network’s usage period.

### 3.1. Energy Harvesting System

The battery is generally used to supply energy for nodes in WSN, but it is necessary that the battery limited energy storage is timely replaced and supplied to ensure the continuous normal operation of the node. In practical application, the real environment is often very complex, which brings great difficulties in laying the power wire or replacing the battery for each node; therefore, it largely limits the universality and flexibility of WSN applications. To solve this problem, we used energy harvesting technology to realize the autonomous power supply of WSN.

At present, the sources of energy harvesting technology generally include solar energy, electromagnetic radiation energy, mechanical vibration energy, and thermal energy. The solar harvesters can obtain several milliwatts of power per square centimeter, which can meet the application demand of wireless sensor nodes. However, the solar cells can convert energy output by the panel only if the light is sufficient, and a single solar energy cannot guarantee a sustainable and reliable power supply for nodes. The energy harvester based on mechanical vibration has received the attention of scholars at home and abroad, because the vibration energy is widely found in the natural environment. The main methods of vibration-to-electric energy conversion are as follows: piezoelectric, electromagnetic, and electrostatic. Among them, the basic principle of piezoelectric vibration energy harvesting is that under the excitation of the external vibration source, the mechanical deformation of the piezoelectric material causes the movement of its internal electrons to generate electric energy. Piezoelectric has the advantages of simple conversion structure, fast response speed, and high energy density, becoming more and more favored by research institutions and high-tech enterprises. The piezoelectric power generation mechanism relies on the d_31_ and d_33_ effects of piezoelectric materials. Among them, the cantilever type piezoelectric generator utilizes the d_31_ effect, which demonstrates a remarkable power output [32,33]. Additionally, the cantilever structure is characterized by its simplicity, high flexibility, and ability to achieve a low natural frequency. Currently, numerous studies [34,35,36] have been conducted on the vibration generation of piezoelectric cantilever beams, with a primary focus on investigating the impact of piezoelectric structure, size, and parameters on power generation. Considering the cost and reliability of the energy harvesting system, this paper organically combines the piezoelectric technology and the solar technology to ensure the energy supply to the sensor nodes.

#### 3.1.1. Overall Scheme

Energy harvesting technology can convert the ubiquitous energy such as thermal energy, solar energy, wind energy, and electromagnetic energy into electrical energy through the appropriate power electronic equipment and the voltage conversion circuit. At present, there are many more developed energy harvesting technologies, and the main source of their energy is generally mechanical vibration energy, electromagnetic radiation energy, solar energy, heat energy, and other energy. In order to realize the self-power supply demand of each sensor node in a WSN, we use solar energy and mechanical energy as the energy source of energy harvesting technology to realize the conversion to electric energy. The framework diagram of node energy supply in a WSN is shown in Figure 2. In this system, the organic combination of piezoelectric power generation technology and solar power generation technology is used to create a complementary effect and fully guarantee the energy supply of the energy harvesting module to the sensor nodes.

#### 3.1.2. Design of Power Supply Circuit

The power supply circuit is mainly divided into three parts. The first is the solar power generation circuit, as shown in Figure 3A. The solar panel can output about 5 V DC voltage, and the large capacitance is used to maintain the stability of the voltage. The second is the piezoelectric ceramic power generation circuit, as shown in Figure 3B. We use the MP1541 chip to design voltage boost conversion circuit with a fixed frequency and peak current mode. The third is the electric energy storage circuit, as shown in Figure 3C. We use the TP4056 chip to design the charging circuit completing the power supply to the lithium battery.

### 3.2. Neighbor Discovery Mechanism

#### 3.2.1. Low-Duty Cycle

In a WSN, nodes must be in a listening state to receive the data from the neighbor node. However, nodes are in idle listening state for most of the time, and the time used for data transmission is often very short, while idle listening causes a lot of invalid energy consumption. Therefore, in many practical applications, nodes adopt low-power communication chips with alternating listening and hibernation modes. These can reduce the unnecessary energy consumption of nodes by replacing idle listening through dormancy.

Considering the application in practice, the low-duty-cycle mechanism is introduced into WSNs, where duty cycle is the ratio of periodic dormancy to activity time, and low-duty cycle means that the duty cycle is not greater than 10%. When the node works in the duty cycle mode, the time will be divided into the time slot of the same size. At a certain number of time slots, the RF will enter the active state that can send and receive data, and the RF will be turned off and will enter the dormant state after the time slot ends. The low-duty-cycle mechanism greatly extends the life cycle of the network and reduces the energy consumption caused by idle listening.

Under the low-duty-cycle mechanism, nodes are in sleep state for most of the time, but nodes are required to open RF as much as possible to communicate with other nodes in order to quickly complete the mutual discovery between nodes. Therefore, saving energy consumption and reducing the detection delay are contradictory, and it is very necessary to study the neighbor discovery problem in the low-duty-cycle wireless sensor networks.

#### 3.2.2. Sending and Receiving Separation Beacon Mechanism Based on Reply to ACK

The neighbor discovery protocol focuses on optimizing the scheduling period when the node works, and it determines the frequency of active time-slot encounters between neighboring nodes, while the beacon mechanism determines the probability of achieving mutual discovery when neighbor nodes meet in an active time slot. At present, most of the studies focus on how to optimize the work scheduling cycle of nodes. There are few studies around the beacon mechanism. In fact, the beacon mechanism also plays a very important role in the discovery of neighbor nodes.

Most of the existing beacon mechanisms in WSN are used for sending and receiving beacons in a single time slot. The sending of the beacon will have certain constraints on the variation range of the time slot size and the listening time in the time slot, which has serious effects on the mutual discovery between nodes, especially when the time slot is small. Therefore, we propose a sending and receiving separation beacon mechanism based on reply to ACK, which only listens to the beacon in the active time slot, and sends the beacon at the end of its previous time slot. In addition, the node immediately replies to ACK after receiving the beacon sent by the adjacent node.

The sending and receiving separation beacon mechanism based on reply ACK is as shown in Figure 4, where nth slot represents the nth time slot, and it is active. At this point, the node will turn on the RF at the end of the (*n* − 1)th slot and transfer the RF from the launch completion state to the send beacon state. Then, the beacon is sent before the end of the (*n* − 1)th slot and the RF is converted to the listening beacon state, and the node will only listen to the beacon in the active time slot. When node A meets node B in an active time slot, node A receives the beacon sent by node B, and it immediately enters the stage of replying to ACK after analysis and verification processing. The bidirectional discovery between nodes can be completed after node B successfully receives ACK.

Discovery probability is an important indicator to measure the performance of the discovery beacon mechanism of neighbor nodes. Improving the discovery probability between nodes can reduce the detection delay and realize the rapid discovery between nodes. When the response ACK mechanism is adopted, the bidirectional discovery can be completed by dynamically extending the time slot after realizing one-way discovery. The bidirectional discovery probability *P_d_* is
(1)$$P_{d}=\frac{t_{slot}-t_{PR}}{t_{slot}-t_{TX}}$$
where *t_slot_* is the time of an active time slot for a node, *t_PR_* is the time that the node takes to send the synchronization packet, and *t_TX_* is the time interval between sending the beacon and listening the beacon.

### 3.3. Data Fusion Mechanism Based on Integer Wavelet Transform

Data fusion is a very important technology in WSN and is a research hotspot. This technology can process a large amount of raw data collected by sensor nodes in various networks through a certain algorithms, remove the redundant information, and transmit only a small number of meaningful processing results to the convergence node. The use of data fusion technology can greatly reduce the amount of data needed to be transmitted in the WSN, reduce data conflicts, reduce network congestion, thus effectively saving energy costs, and prolong the life of the network.

In WSN, the perceptual data collected by nodes from the environment are time-series data, as a strong time correlation and periodicity because of the short interval, while the dataset generated by the whole network has a certain spatial correlation due to the close distance. For spatial correlation between data, integer wavelet transform can also be used to remove redundant parts.

#### 3.3.1. Integer Wavelet Transform

Spatiotemporal data have strong linear features that consist of multiple curves and are well suited for applying the wavelet transform. After wavelet transformation, most of the energy of the data is stored in the low-frequency component, and the detail information is stored in the high-frequency component. It does well in removing the redundant information from the original data.

The integer wavelet transform is a new wavelet construction method and has the following advantages: (1) It has only the integer shift and addition and subtraction operation, processes data quickly, has low hardware requirements, and is easy to implement. (2) It is completely reversible, which can perform both lossy coding and lossless coding. There are 12 commonly used integer wavelet filters, which have different computational complexity and compression performance. Among them, 5/3 wavelet has a small calculation amount and a good compression effect, which is often used to perform lossless compression of spatiotemporal data [37]. Its transformation formula is as follows:(2)$$\begin{array}{l} d_{1,j}=s_{0,2i+\_1}-\left\lfloor \frac{s_{0,2i}+s_{0,2i+2}}{2}\right\rfloor \\ s_{1,j}=s_{0,2i}+\left\lfloor \frac{d_{1,j-1}+d_{1,j+2}}{4}\right\rfloor \end{array}$$
where *s_i,j_* is the *j*-th approximation of the *k*-level transformation decomposition, *d_i,j_* is the *j*-th detail value of the k-level transformation decomposition, and ⌊·⌋ means rounding down. In practice, we need to use symmetric periodic extension for boundary data.

#### 3.3.2. Select Sink Node and Construct Hierarchical Structure

(1)Select sink node

The Sink node undertakes a lot of operations of data processing and forwarding within the group. Often, the node with strong ability is selected as the Sink to complete the communication function of the group and extend the life cycle of the network at the same time. When the Sink node is selected, three factors need to be considered: residual energy, link quality, and degree. Therefore, we set a weight *w*, and then the Sink is finally determined based on the calculated *w* value. The calculation of *w* can be defined as follows:(3)$$w_{i}=\alpha \cdot \frac{E_{i}}{\frac{1}{n}\sum_{j=1}^{n}E_{j}}\cdot \frac{E_{i}}{E_{0}}+\beta \cdot \frac{\sum_{j=1}^{n}L_{ij}}{L_{\max}}+\gamma \cdot \frac{D_{i}}{D_{a}}$$
where *E_i_* is the residual energy of the node *v_i_*, *E_0_* is the initial energy of nodes, *E_j_* is the residual energy of the neighbor node, *L_ij_* is the link quality between the node *v_i_* and the neighbor node, *L_max_* is the maximum value of link quality in the group, *D_i_* is the degree of node *v_i_*, *D_a_* is the average of degrees of nodes in the group, and *α*, *β*, *γ* is the adjust parameters, and satisfy *α* + *β* + *γ* = 1.

(2)Construct hierarchical network structure

The nodes in the group transfer the data to the convergence nodes by a hop transmission, and then the convergence nodes should forward these data to the Sink node. To achieve this submission mode, it is necessary to build a hierarchical network structure.

As shown in Figure 5, the white nodes are the ordinary nodes in the network, and they can be layered by the message flooding of Sink nodes. In this process, the Sink node broadcasts a request message to all adjacent nodes, and these nodes that have received the message need to set their layer value to 1 and continue to broadcast the message to the surrounding nodes. After receiving the message, the surrounding nodes should first check whether their layer value has been set. If not, their own layer value adds 1 on the basis of the broadcast layer value; that is, their layer value is 2. These nodes with level 2 continue to broadcast this message; thus, the third layer node will be formed, and then all nodes will finally have their own layer value.

#### 3.3.3. Data Fusion Based on Integer Wavelet Transform

The data collected by a single node have strong temporal correlation due to the short interval, while the data generated by multiple nodes in the network have some spatial correlation due to their close distance. We used integer wavelet transform techniques to remove spatiotemporal dependent redundancy of these data.

(1)Time-dimensional data fusion of single node

The operation process of time-dimensional data fusion using integer wavelet transform is as follows. Step 1: Data preprocessing. It mainly carries out data cleaning and data integer transformation on the raw data collected by the nodes. Step 2: Integer wavelet transform. The time-dimensional data series is decomposed into low- and high-frequency coefficients based on integer wavelet transform. Step 3: Coefficients quantification. The wavelet coefficients were quantified by using scalar quantification techniques. Step 4: Encoding. The wavelet coefficients are encoded using the encoding algorithm, such as the deflate algorithm.

(2)Spatial–dimensional data fusion of nodes

The operation process of spatial–dimensional data fusion using integer wavelet transform is as follows. Step 1: Integer wavelet transformation of data for the underlying node. The nodes underlying the network are divided into the even nodes *s_0_*_,*i*_ and the odd nodes *d*_0*,i*_ in the order, and then the even node *s*_0,1_ transmits its data to the adjacent odd node *d*_0,1_ and calculates the integer wavelet coefficients *s*_1,1_ and *d*_1,1_. Correspondingly, all the coefficients of the one-level wavelet decomposition (*d*_1,1_, *d*_1,2_, …, *d*_1,*n*_) and (*s*_1,1_, *s*_1,2_, …, *s*_1,*n*_) can be obtained. Step 2: Integer wavelet transform of data for the convergence node. The underlying odd nodes send the low-frequency coefficient (*s*_1,1_, *s*_1,2_, …, *s*_1,*n*_) to the convergence nodes of the upper layer and form a new data sequences. By repeating the Step 1 operation, we can obtain the three-level wavelet transform coefficients (*d*_2,1_, *d*_2,2_, …, *d*_2,*n*_) and (*s*_2,1_, *s*_2,2_, …, *s*_2,*n*_). Step 3: Integer wavelet transform of data for the Sink node. The Sink node obtains the k-level wavelet transform coefficients (*d_k_*_,1_, *d_k_*_,2_, …, *d_k_*_,*n*_) and (*s_k_*_,1_, *s_k_*_,2_, …, *s_k_*_,*n*_). The diagram of spatial–dimensional data fusion of nodes is shown in Figure 6.

## 4. Simulations and Results

### 4.1. Energy Harvesting System

#### 4.1.1. Performance of the Solar Energy Harvester

The test data of the energy harvesting system that we designed are shown in Table 1, and the table records the test data of this system on a sunny day in Yiyang, Hunan Province. The elements of the test mainly include voltage regulator circuit input voltage, node power supply voltage, and node power supply current. The test results show that the energy harvesting system is stable in power supply to the node, maintains about 3.3 V, has less than 1.3% in voltage deviation, and it has high accuracy, strong robustness, and can provide reliable energy guarantee for WSNs.

#### 4.1.2. Performance of the Bistable Piezoelectric Cantilever Oscillator

(1)Simple and harmonic incentive

In this experiment, we examined the power generation performance of a bistable piezoelectric cantilever oscillator composed of two magnets under simple harmonic incentive, with varying magnetization strength and load resistance. In Figure 7A, the voltage amplitude frequency curve is compared for different magnetization intensities. The system’s voltage amplitude solution exhibits jumps as the excitation frequency changes. Furthermore, an increase in the nonlinear term coefficient results in a rightward shift of the curve, widening the response band of large motion and increasing the amplitude. Figure 7B displays the power generated at different impedances. It is noteworthy that the jump phenomenon’s corresponding excitation frequency in the system’s amplitude and frequency curve remains unchanged regardless of the resistance value. In the system, there is an issue with resistance matching. Specifically, the resistance value R = 30 k needs to be addressed and optimized.

(2)Random incentive

In this experiment, we examined the impact of various incentive strengths and load resistances on the response of a bistable piezoelectric cantilever subjected to random incentive. The excitation strength was set to 0.01~0.09 G2/Hz and the load resistance was 1 k. As shown in Figure 8A, the output voltage of the system increases proportionally with the excitation strength. The incentive strength is 0.06 G2/Hz, and Figure 8B shows the average power curve when the impedance changes. The figure clearly demonstrates that at a resistance of 19.1 k, the output power reaches its peak. This signifies that matching the impedance maximizes the output power of the system when subjected to random excitation.

### 4.2. Neighbor Discovery

In this experiment, the sending and receiving separation beacon mechanism based on reply to ACK (SRSB-A) is tested on the operating system Tiny OS 2.0 and the sensor node Telos B. To evaluate the performance of SRSB-A, we compare it with protocols such as Disco, U-Connect, etc. Figure 9A shows the contrast experiment with a 5% duty cycle in asynchronous symmetric scenes. The results show that the node in the SRSB-A can be successfully found by its neighbor node within 320 time slots, and in the worst discovery delay, the SRSB-A was reduced by 66.67% and 75% over Disco and U-connect, respectively. As shown in Figure 9B, in the contrast experiment with a 5% duty cycle in asynchronous asymmetric scenes, the node in the SRSB-A can successfully discover neighbor nodes within 3600 time slots. In the worst discovery delay, the SRSB-A was reduced by 53.37% and 71.27% over Disco and U-connect, respectively. Based on the above analysis, it can be considered that the SRSB-A has a good performance.

### 4.3. Data Fusion

In this experiment, the experimental data were derived from the Tropical Atmospheric Ocean Project (TAO). Since 1984, the project has deployed about 100 sensors at different depths in 71 moorings in the tropical Pacific to collect the temperature of seawater in the region. We used these data for the lossless compression testing, and the test computer configuration was as follows: Processor: Intel^®^ Core™ i5-6500 CPU @ 3.20 GHz (Intel, Santa Clara, CA, USA), memory: 16 GB, operating system: 64-bit operating system. The test results are shown in Table 2. Compressed by using integer wavelet transform, the spatial and temporal data can obtain an average compression rate of 5.42. Compared with other lossless compression algorithms Huffman, LZSS, LZW, and WinRAR, the integer wavelet transform has a higher compression ratio, more than 1 times higher.

On the other hand, we took the raw data of 96 sensors from the dataset of TAO to carry out the three-level integer wavelet transform, and then analyzed the data storage of nodes. After integer wavelet transform, the one-level wavelet coefficients were (*s*_1,1_, *s*_1,2_, …, *s*_1,52_) and (*d*_1,1_, *d*_1,2_, …, *d*_1,52_), the two-level wavelet coefficients were (*s*_2,1_, *s*_2,2_, …, *s*_2,30_) and (*d*_2,1_, *d*_2,2_, …, *d*_2,30_), and the three-level wavelet coefficients were (*s*_3,1_, *s*_3,2_, …, *s*_3,19_) and (*d*_3,1_, *d*_3,2_, …, *d*_3,19_). Among these data, the data (*d*_1,1_, *d*_1,2_, …, *d*_1,52_) were stored in the bottom node, the data (*d*_2,1_, *d*_2,2_, …, *d*_2,30_) were stored in the convergence node, and the data (*d*_3,1_, *d*_3,2_, …, *d*_3,19_) were stored in the Sink node. Compared with the traditional communication mode, the amount of data stored in the integer wavelet transform was much smaller, as shown in Table 3, which indicates that the integer wavelet transform has certain advantages in reducing the amount of communication data and transmission energy.

## 5. Discussion

For WSNs with limited energy, the energy management of nodes is still one of the most important factors restricting their large-scale and long-term deployment and application.

(1)Energy harvesting technology

It is an effective power supply technology for nodes to harvest the energy produced by the surrounding environment. In the energy storage module, we use capacitors to store energy, which can avoid the problems of node failure caused by frequent charging and discharge of batteries, but the capacitors have the shortcomings of less storage energy and large leakage current. In addition, the energy collection, storage, and management circuit will also consume electricity so that the utilization rate of energy is not high, and only when the solar panel voltage is higher than the capacitor voltage is the solar energy used by the system.

(2)Neighbor discovery mechanism

Low-duty cycle can reduce the idle listening time and save a lot of energy. However, the current mechanism of neighbor beacon discovery based on low-duty cycle is too idealistic in analyzing the discovery probability and beacon conflict between nodes, which is quite different from the actual working situation between nodes. In addition, the factors such as the size of time slot and the number of beacons in the active time slot will have an important impact on the performance of the neighbor discovery. In this paper, our time slot model is closer to the actual working process of sensor nodes, and the sending and receiving separation beacon mechanism based on reply to ACK can effectively reduce the beacon conflict. In addition, the low-duty cycle mechanism can result in longer detection delays. In general, longer detection delays can miss some communication opportunities, which is not acceptable. However, shorter detection delays can consume more energy; thus, a balance is required between the detection delay and energy expenditure in the application of WSNs.

(3)Data fusion mechanism based on integer wavelet transform

Data fusion can greatly reduce the amount of data in the network, which is conducive to reducing the energy expenditure of nodes by data transmission. Lossless data compression based on the integer wavelet transform can reach a compression ratio of 4–19 times, which can better remove the spatial and temporal correlation of the data. In addition, the distributed preservation of multiresolution data can solve the limitation of the storage space of nodes and improve the disaster recovery capacity of the network. However, the integer wavelet transform has some requirements on the computational power of the nodes.

## 6. Conclusions

In this paper, we discussed the problem of node energy management in wireless sensor networks. In order to extend the service life of the network, we took three measures. Firstly, the solar panels and pressure ceramics were used to harvest solar energy and vibrating mechanical energy in nature to supply energy to the sensor nodes. Secondly, the neighbor discovery mechanism based on low-duty cycle was studied to reduce the idle listening energy consumption of nodes. Finally, the data fusion mechanism based on the integer wavelet transform was studied to reduce the transmission data energy consumption of the node. The experimental results show that these measures are effective in reducing the energy consumption of nodes and extending the life span of the network. However, there are numerous challenges and opportunities that still exist in the field of energy management in WSNs. Further research should focus on developing more efficient energy collection and conversion technologies, establishing dynamic models of bistable piezoelectric generation systems, and exploring the response characteristics of the latter under random excitation. Additionally, optimizing the design of bistable piezoelectric generation systems is crucial. Furthermore, the low power technology of nodes needs to be further studied in order to design and optimize the energy management system for the entire wireless sensor network. This includes node energy management, energy distribution strategies, and energy balance technologies. By effectively scheduling the energy usage of the nodes, the stable operation and long lifespan of the entire network can be ensured.

## Figures and Tables

**Figure 1 sensors-24-00047-f001:**
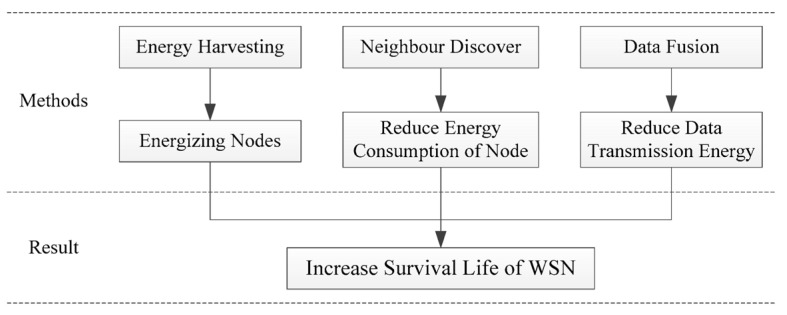
Framework diagram of technical route in a WSN.

**Figure 2 sensors-24-00047-f002:**
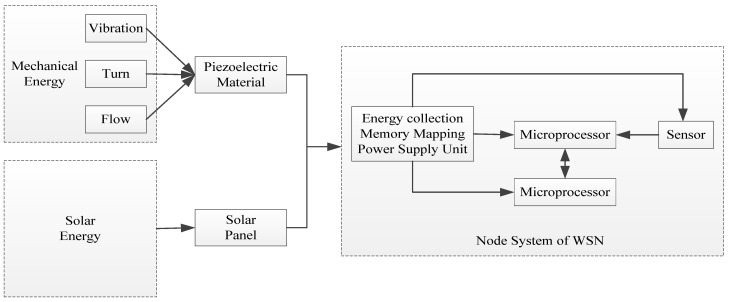
Framework diagram of node energy supply in a WSN.

**Figure 3 sensors-24-00047-f003:**
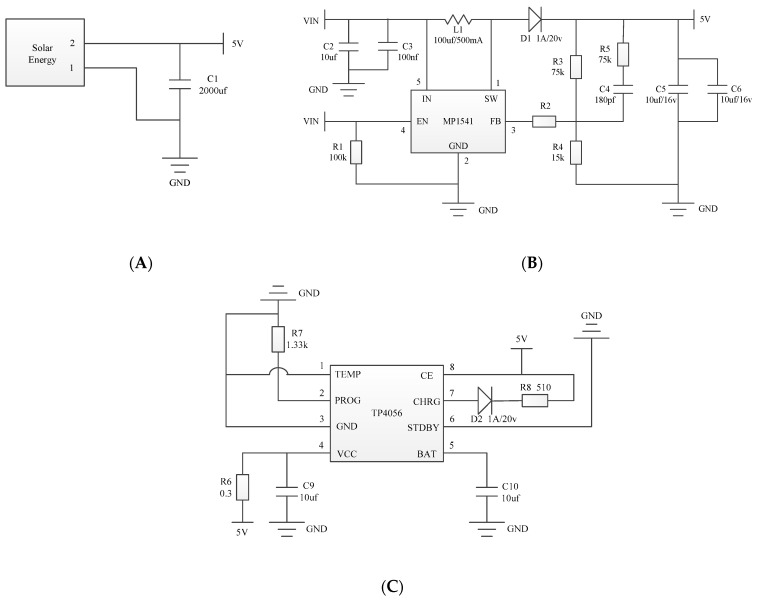
Diagram of power supply module: (**A**) solar power generation circuit, (**B**) piezoelectric ceramic power generation circuit, and (**C**) electric energy storage circuit.

**Figure 4 sensors-24-00047-f004:**
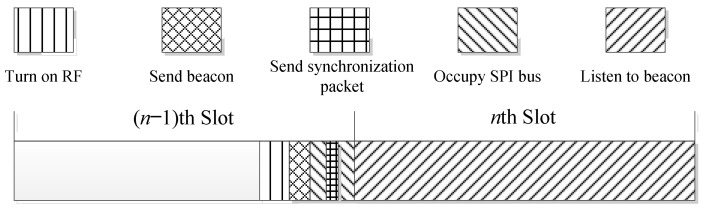
Time slot model of sending and receiving separation beacon mechanism based on reply ACK.

**Figure 5 sensors-24-00047-f005:**
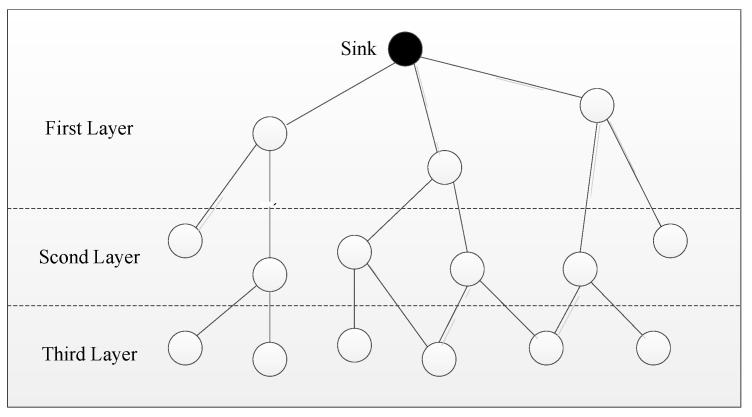
Diagram of hierarchical network structure in a WSN.

**Figure 6 sensors-24-00047-f006:**
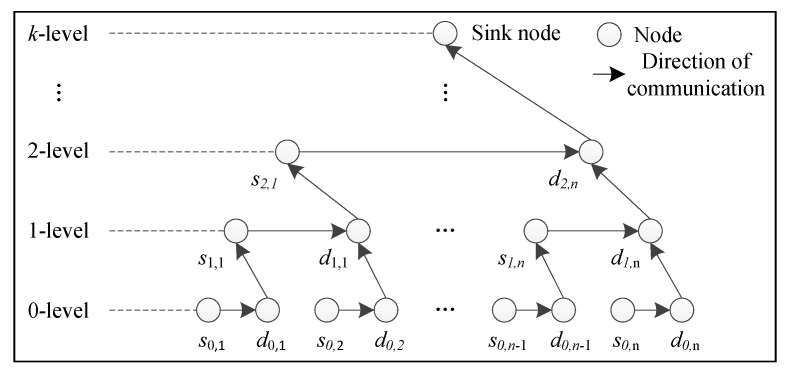
Diagram of spatial–dimensional data fusion of nodes.

**Figure 7 sensors-24-00047-f007:**
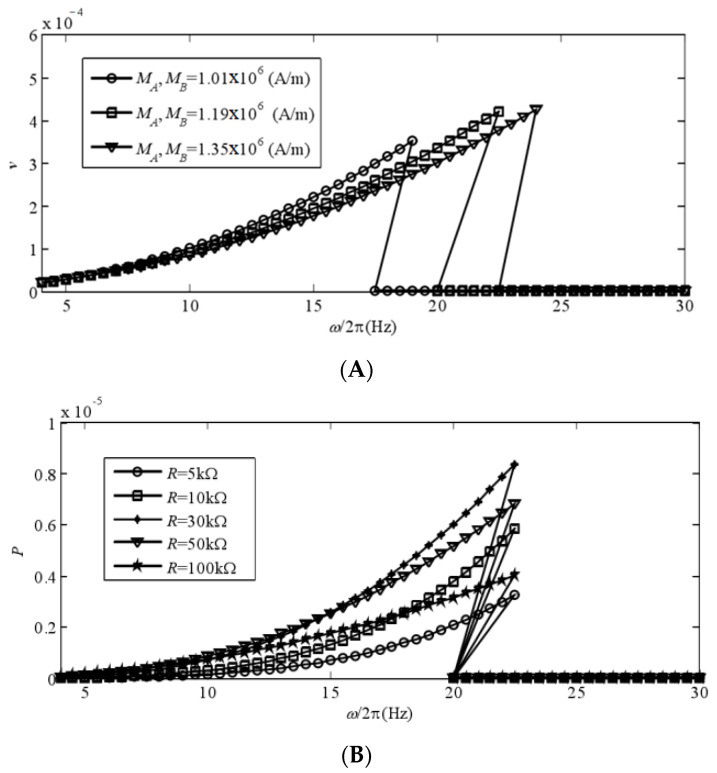
The power generation performance of a bistable piezoelectric cantilever oscillator: (**A**) amplitude–frequency curves at different magnetization strengths; (**B**) output power at different impedance levels.

**Figure 8 sensors-24-00047-f008:**
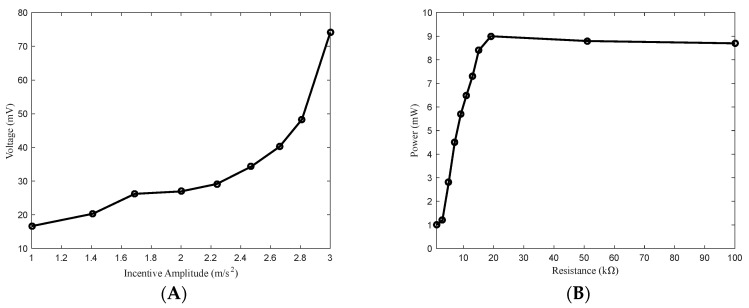
The power generation performance of a bistable piezoelectric cantilever oscillator: (**A**) average voltage curves for different excitation intensities; (**B**) average power curves at the different impedances.

**Figure 9 sensors-24-00047-f009:**
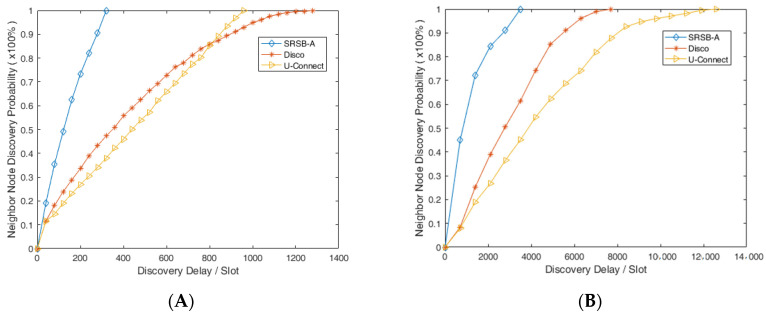
Contrast experiment for neighbor node discovery: (**A**) symmetry scene and (**B**) asymmetric scene.

**Table 1 sensors-24-00047-t001:** Measured data of energy harvesting system.

Time	Circuit Input Average Voltage	Power Supply Average Voltage	Power Supply Average Current	Power Supply Voltage Deviation
6:00	5.00 V	3.33 V	100.0 mA	0.91%
9:00	4.97 V	3.30 V	99.8 mA	0%
12:00	4.97 V	3.29 V	102.6 mA	0.30%
15:00	4.98 V	3.31 V	101.4 mA	0.30%
18:00	4.21 V	3.28 V	99.9 mA	0.61%
21:00	4.19 V	3.27 V	100.0 mA	0.91%
24:00	4.99 V	3.32 V	100.1 mA	0.61%
3:00	5.02 V	3.34 V	103.4 mA	1.21%

**Table 2 sensors-24-00047-t002:** Compression ratio of different lossless compression algorithms.

Algorithm	File Size (Byte)	Compression Time (ms)	Uncompressing Time (ms)	Compression Ratio
Integral Wavelet	151,982	32	15	5.42
Huffman	151,982	44	18	2.05
LZSS	151,982	62	37	2.04
LZW	151,982	49	25	2.36
WinRAR	151,982	31	16	2.52

**Table 3 sensors-24-00047-t003:** Comparison of the data traffic.

Mode	Number of Data Stored		
	Bottom Node	Convergence Node	Sink Node
Traditional Model	96	96	96
Integral Wavelet	52	30	19

## Data Availability

Data are contained within the article.

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
