# Peer review of "Research on Energy Harvesting Mechanism and Low Power Technology in Wireless Sensor Networks"

_sensors, 2023, doi:10.3390/s24010047_

Round 1

Reviewer 1 Report (Previous Reviewer 4)

Comments and Suggestions for Authors

This paper investigates the energy harvesting mechanism and low-power technology of wireless sensor networks. The studied topic is important, and the authors have resubmitted the manuscript after addressing the major comments earlier. 

I believe substantial changes have been made compared to an earlier version. Therefore, I have only a few minor comments to be addressed:

- Typo in the abstract, line 7, "Therefor" -> "Therefore".

- I suggest including subsection headers under related work for the main three areas described ( i.e., 2.1. Energy harvesting technology, 2.2 Low duty cycle working mode and 2.3. Data fusion mechanism).

- Figure 8B, please improve the resolution of the figure, as the resolution seems different than Figure8A. On top of that, please capitalize "Voltage" in the axis name in Figure 8A.

- The size of Figure 7A and Figure 7B need to be increased. If possible, they can be adjusted side by side.

Besides these minor improvements, the content provided in the methodology and the rest of the sections are sufficient and comprehensive.

Comments on the Quality of English Language

The level of English is satisfactory.

Author Response

Dear Reviewer:

We are very grateful to your comments for the manuscript. According with your advice, we amended the relevant part in manuscript. Some of your questions were answered below.

1. Typo in the abstract, line 7, "Therefor" -> "Therefore".

Response: We were really sorry for our careless mistakes. This kind of problem has been checked and corrected.

2. I suggest including subsection headers under related work for the main three areas described ( i.e., 2.1. Energy harvesting technology, 2.2 Low duty cycle working mode and 2.3. Data fusion mechanism).

Response: Thank you for pointing this out. We have revised it in the revised manuscript.

 3. Figure 8B, please improve the resolution of the figure, as the resolution seems different than Figure8A. On top of that, please capitalize "Voltage" in the axis name in Figure 8A.

Response: As the Reviewer's good advice, we have revised it in the revised manuscript.

4. The size of Figure 7A and Figure 7B need to be increased. If possible, they can be adjusted side by side.

Response: As the Reviewer's good advice, we have revised it in the revised manuscript.

Reviewer 2 Report (New Reviewer)

Comments and Suggestions for Authors

This article delves into the energy collection technology of WSN, explores the use of solar energy and mechanical vibration energy to ensure continuous and reliable power supply of sensor nodes, and analyzes the voltage output characteristics of dual voltage stabilized cantilever beams. Three measures: energy harvesting system, neighbor discover mechanism, and data fusion mechanism have been taken to address the issue of node energy management in wireless sensor networks. The construction and simulation of the overall model are complete. The experimental results indicate that these measures are effective in reducing node energy consumption and extending network lifespan. In my opinion, this work falls within the scope of Sensors and should be published with slight modifications. The following are several suggestions and explanations that must be addressed before publication.

1.        The authors state “Discovery probability is an important indicator to measure the performance of the discovery beacon mechanism of neighbor nodes.” It is best to use some data or graphical curves to demonstrate the improvement of discovery probability between nodes.

2.        The description of the WSN node system in the overall framework diagram of Figure 2 is not detailed enough.

3.        The authors state “However, under the low duty cycle mechanism, nodes are in sleep state for most of the time...” Saving energy consumption and reducing the detection delay are contradictory. The authors need to explain how to strike a balance between detection delay and energy consumption in the transceiver separation beacon mechanism based on ACK response?

4.        The layout of Figure 7 needs to be re planned.

5.        The presented design has good potentials on intelligent sensing, could the author provide some discussion. The following paper could be included: doi:10.1002/smll.202203871; doi: 10.1038/s41928-023-00951-x.

Author Response

Dear Reviewer:

We are very grateful to your comments for the manuscript. According with your advice, we amended the relevant part in manuscript. Some of your questions were answered below.

1. The authors state “Discovery probability is an important indicator to measure the performance of the discovery beacon mechanism of neighbor nodes.” It is best to use some data or graphical curves to demonstrate the improvement of discovery probability between nodes.

Response: Thank you for making a very good point here. In this paper, we mainly consider the comprehensive use of the technologies (such as energy harvesting, neighbor discovery and data fusion) to analyze the energy supply and energy consumption of nodes in WSN. Therefore, we did not introduce and analyze in detail the discovery probabilities between nodes , but only contrast with some classical methods. However, we will further study the operation, discovery performance and versatility of the mechanism.

2. The description of the WSN node system in the overall framework diagram of Figure 2 is not detailed enough.

Response: Thank you for pointing this out. We have revised it in the revised manuscript.

3. The authors state “However, under the low duty cycle mechanism, nodes are in sleep state for most of the time...” Saving energy consumption and reducing the detection delay are contradictory. The authors need to explain how to strike a balance between detection delay and energy consumption in the transceiver separation beacon mechanism based on ACK response?

Response: Thank you for your guidance. Saving energy consumption and reducing the detection delay are contradictory. In order to strike a balance between detection delay and energy consumption, we used two approaches: reply based on ACK and random retreat. Without adding additional energy overhead, the mechanism based on the reply ACK can reduce the beacon conflict between nodes, improve the probability of bidirectional discovery, have a lower detection delay. At the same time, the random retreat mechanism can alleviate the continuous conflict problem between nodes and reduce the maximum detection delay between nodes.

4. The layout of Figure 7 needs to be re planned.

Response: As the Reviewer's good advice, we have revised it in the revised manuscript.

5. The presented design has good potentials on intelligent sensing, could the author provide some discussion. The following paper could be included: doi:10.1002/smll.202203871; doi: 10.1038/s41928-023-00951-x.

Response: Thank you for pointing this out. We have revised it in the revised manuscript.

This manuscript is a resubmission of an earlier submission. The following is a list of the peer review reports and author responses from that submission.

Round 1

Reviewer 1 Report

Comments and Suggestions for Authors This paper presented a solution to the WSN's power, receiving and sending mechanism and data fusion.  The energy harvesting system can provide reliable power for WSN by effectively harvesting solar energy and mechanical energy in the ambient environment. The neighbor discovery mechanism of sending and receiving separation beacon based on response to ACK enables nodes to find their neighbor soon. The data fusion mechanism based on integer wavelet transform is able to effectively remove the data redundancy caused spatio-temporal correlation and reduces the energy consumption of nodes for data transmission.  1) In section 3.1, the introduction about energy harvesting is brief, some detailed information should be provided, e.g., the introduction about vibration energy harvesting, how much is the output? 2) Section 4 is "Experimental Simulation and Results", to my understanding, there is simulation but no experiment in this paper, so the title should be "Simulations and results" 3) In Sections 4.1 ,4.2 and 4.3, the titles have different uppercase letters.

Comments on the Quality of English Language

The English should be improved.

Author Response

I am very grateful to your comments for the manuscript. According with your advice, we amended the relevant part in manuscript. Some of your questions were answered below.

1.In section 3.1, the introduction about energy harvesting is brief, some detailed information should be provided, e.g., the introduction about vibration energy harvesting, how much is the output?
Response: Thank you for your guidance. The section has been rewrote in the revised manuscript. 

2.Section 4 is "Experimental Simulation and Results", to my understanding, there is simulation but no experiment in this paper, so the title should be "Simulations and results".
Response: Thank you for pointing this out. We have revised it in the revised manuscript. 

3.In Sections 4.1 ,4.2 and 4.3, the titles have different uppercase letters.
Response: We were sorry for our careless mistakes. This kind of problem have been checked and corrected.

Reviewer 2 Report

Comments and Suggestions for Authors

1. Better to explain the technical content in place of general content here: "WSN by effectively collecting solar energy and mechanical energy in the ambient environment. The neighbor discovery mechanism of sending and receiving separation beacon based on reply to ACK enables nodes to find their neighbor at a certain duty cycle ratio, and shortens the idle listening time and reduces a lot of energy consumption"

2. Introduction section is very weak. Try to justify the problem defination, motivation and contribution by citing appropriate work

3. Litrature table is required for comparison with limitations and proposed work

4. Figure  and 6 depicts that work is as per the following paper: 

1. Energy efficient chain based cooperative routing protocol for WSN

2. A novel scheme for an energy efficient Internet of Things based on wireless sensor networks

However, work is not compared with this. Whereas the architecture is the same for nodes communication

6. Overall manuscript needs major changes and on novelty part it is very weak

Author Response

I am very grateful to your comments for the manuscript. According with your advice, we amended the relevant part in manuscript. Some of your questions were answered below.

1.Better to explain the technical content in place of general content here: "WSN by effectively collecting solar energy and mechanical energy in the ambient environment. The neighbor discovery mechanism of sending and receiving separation beacon based on reply to ACK enables nodes to find their neighbor at a certain duty cycle ratio, and shortens the idle listening time and reduces a lot of energy consumption".
Response: Thank you for your guidance. These content has been rewrote in the revised manuscript.

2.Introduction section is very weak. Try to justify the problem defination, motivation and contribution by citing appropriate work.
Response: Thank you for pointing this out. Introduction section has been revised. 

3.Litrature table is required for comparison with limitations and proposed work.
Response: Thank you for your suggestions. We have carefully revised the table.

4.Figure 5 and 6 depicts that work is as per the following paper: 1. Energy efficient chain based cooperative routing protocol for WSN. 2. A novel scheme for an energy efficient Internet of Things based on wireless sensor networks. However, work is not compared with this. Whereas the architecture is the same for nodes communication.
Response: Thank you for pointing this out. Figure 5 and 6 are used to explain the data fusion based on integer wavelet transform. After the data of adjacent nodes is transformed in integer wavelet, only the low frequency coefficient is sent to the upper layer nodes. Clustering transmission is a common method for wireless sensor networks, but in this paper, it can also realize multi-resolution storage of data.

5.Overall manuscript needs major changes and on novelty part it is very weak.
Response: We are sorry. According to your advice, we rewritten the Abstract, Introduction, 3.1. Energy Harvesting System, 4 Simulations and Results, and so on.

Reviewer 3 Report

Comments and Suggestions for Authors

The presented contribution summarizes current research on wireless sensor networks and provides a brief overview of key research areas, such as energy harvesting, working modes and data transmission, and data fusion. However, in some of these areas are not broke down in necessary detail and are too general. Related work (chapter 2), especially the part confronting wireless charging technology, seems incomplete and without any conclusions in relation to the presented work. Authors did not comment on their choice of using piezoelectric material for vibration energy harvesting, considering there are many other VEH technologies, many of which are superior in terms of generated power density and efficiency. It is not described what kind of data the individual nodes collect and whether the presented data fusion techniques are suitable only for a specific type of data, especially in terms of present frequencies, length of measured data and periodicity. In chapter 4.3, only data from long-term monitoring of temperature was used, therefore authors should comment on feasibility of the presented approach for signals with shorter time constant, such as vibrations. Furthermore, how is it determined whether the collected raw data contains any redundant information to be removed?

There is no information about the energy harvesting system used in experiment, results of which are presented in Table 1. Was that only a solar panel or also a vibration energy harvesting system consisting of the piezoelectric harvester previously mentioned and depicted in Fig. 2? If it is the latter case, what were the parameters of the vibrations? I am also missing an energy / power breakdown of individual duty cycles and transmission / compression techniques, in other words, how much power the system needs at any time (data acquisition, preprocessing and compression, transmission, storage etc.).

Besides the abovementioned issues, the presented contribution has many formatting issues. In the presented form, I do not believe the contribution has high enough scientific value and the presented techniques are too general without any specific conclusions or applicability for current WSN systems. I recommend acceptance only after a major revision.

Comments on the Quality of English Language

Besides many formal formatting issues, technical English is at an acceptable level, but the submitted contribution contains a number of grammatical errors which should be addressed.

Author Response

I am very grateful to your comments for the manuscript. According with your advice, we amended the relevant part in manuscript. Some of your questions were answered below.

1.in some of these areas are not broke down in necessary detail and are too general. There is no information about the energy harvesting system used in experiment, Besides the above mentioned issues, the presented contribution has many formatting issues. 
Response: Thank you for your valuable suggestions, and please allow me to answer it together. According to your advice, we rewritten the Abstract, Introduction, 3.1. Energy Harvesting System, 4 Simulations and Results, and so on. In addition, we have checked and corrected the manuscript. The above revision may not be perfect due to the time constraints, thank you for your comments.

Reviewer 4 Report

Comments and Suggestions for Authors

MDPI Sensors Journal (Manuscript ID: sensors-2484463)

Comments to the Author

This paper proposes an energy harvesting system and low energy consumption mechanism in wireless sensor networks. The paper studies a useful topic. However, there are several points that need to be addressed to improve the quality of the manuscript.

Suggestions to improve the quality of the paper are provided below:

1)     Please indicate the corresponding author’s institutional email address after author affiliations.

2)     The flow in the abstract should be improved. For example, the authors emphasize that finite power energy is an important constraint in one sentence and the next one is the objective is addressing the contradiction between an extended life cycle and the energy expenditure of sensor nodes. How these two relate should be better connected to make the motivation more coherent.

3)     Similar to my comment above, in order to improve the abstract, please use past tense to explain the methodology and the work conducted. This part should explain what was done in this research. The sentence indicating “The energy harvesting system can provide reliable energy guarantee …” does not sound like you’re describing what has been done in this work but it reads as general information about energy harvesting systems. Please restructure the sentences in this direction, and highlight the important findings and implications of this work as an ending sentence.

4)     Please clearly state the novelty and contributions of this work in bullet points, and highlight the ways in which it improves upon the previous works in the introduction.

5)     Another important observation is that the introduction is very short, providing almost no reference (currently only one in the entire introduction) and definitely needs to be expanded with a broader overview as a starting paragraph. This part can be expanded with of “IoT/AI and wireless sensor networks” providing a general perspective in the first paragraph of the Introduction.  On top of that several popular application areas/fields should be mentioned with supporting literature to emphasize in this paragraph. Some examples of IoT & WSN integration include healthcare [https://ieeexplore.ieee.org/document/5570866], appliance energy management [https://doi.org/10.1016/j.buildenv.2022.109472] , smart transportation [https://ieeexplore.ieee.org/document/8984291], thermal comfort [https://doi.org/10.1016/j.buildenv.2023.110148] and energy load identification [https://doi.org/10.1016/j.apenergy.2020.115391].

Please review these applications as a good starting point, as part of the general applications of “IoT/AI and wireless sensor networks” in the introduction and provide a broader overview to improve the quality of the introduction. (this should come before “the nodes of WSN” paragraph in the existing manuscript)

6)     A low-duty cycle mechanism was introduced to reduce the energy consumption caused by idle listening, resulting in longer detection delays. Please discuss about the implications of longer detection delays and also highlight some scenarios where longer detection delays are acceptable/unacceptable.

7)     In section 3.3.1, the authors made the statement “Among them, 5 / 3 wavelet has a small calculation amount and a good compression effect, which is often used to do lossless compression of spatiotemporal data.” but did not provide any references. Please include some references to support this statement.

8)     In Section 4.3, please provide some additional details about the experiment conducted. What is the specifications of the machine used to perform the compression? How many times was the compassion experiment repeated? While the algorithms were stated to be lossless, was this verified during the experiment?

9)     It is great that the authors have spent some time thinking about the limitations of the proposed approach in the Discussion section. However, the authors should also think about what can be explored in the future to address those limitations.

10)  Comments on paper structure and writing:

·       Please change real time to “real-time” in the overall manuscript.

·       Please avoid using “Literature[12]” when you give a reference, instead use the author's name such as “XX et al [12].” This applies to all literature following this form.

·       Section 3.2. should start with capital letters in each word (i.e., Neighbor Discovery Mechanism”.

·       Same comment above applies to Section 3.3 and 4.2, 4.3. Please be consistent with the section structures.

·       The information in Table 3 seems to be cut off for the first column.

Comments on the Quality of English Language

The level of English can be improved by addressing the coherence issues, especially in the abstract.

Author Response

I am very grateful to your comments for the manuscript. According with your advice, we amended the relevant part in manuscript. Some of your questions were answered below.

1.Please indicate the corresponding author’s institutional email address after author affiliations.
Response: Thank you for your suggestions. the corresponding author’s institutional email address is fangtang@scut.edu.cn. 

2.The flow in the abstract should be improved. please use past tense to explain the methodology and the work conducted. Please clearly state the novelty and contributions of this work in bullet points.
Response: Thank you for pointing this out. The abstract has been revised.

3.Another important observation is that the introduction is very short, providing almost no reference (currently only one in the entire introduction) and definitely needs to be expanded with a broader overview as a starting paragraph. 
Response: Thank you for your suggestions. We have revised the introduction.

4.A low-duty cycle mechanism was introduced to reduce the energy consumption caused by idle listening, resulting in longer detection delays. Please discuss about the implications of longer detection delays and also highlight some scenarios where longer detection delays are acceptable/unacceptable.
Response: Thank you for your suggestions. We add the discuss into Discussion section. 

5.In section 3.3.1, the authors made the statement “Among them, 5 / 3 wavelet has a small calculation amount and a good compression effect, which is often used to do lossless compression of spatiotemporal data.”but did not provide any references. Please include some references to support this statement.
Response: Thank you for your suggestions. We have revised the references.

6. In Section 4.3, please provide some additional details about the experiment conducted. What is the specifications of the machine used to perform the compression? How many times was the compassion experiment repeated? While the algorithms were stated to be lossless, was this verified during the experiment?
Response: Thank you for pointing this out. We have added the test computer configuration. the compassion experiment were performed multiple times, and the results are averaged. In addition, the algorithm is a theoretically lossless compression algorithm, but the compressed data must be integer (real data can be transformed into integer through unit conversion). The data in this experiment is temperature value, it is easy to transform into integers, so lossless compression can be achieved. In other scenarios, the data is too costly to be an integer and may require lossy compression.

7. It is great that the authors have spent some time thinking about the limitations of the proposed approach in the Discussion section. However, the authors should also think about what can be explored in the future to address those limitations.
Response: Thank you for your suggestions. In fact, we are thinking about these question, but there is no better way to address those limitations.

8. Comments on paper structure and writing.
Response: We are sorry. We have revised these errors.

Round 2

Reviewer 2 Report

Comments and Suggestions for Authors

1.     Technically, the paper is well written.

2.     Researchers have put forward an effective strategy on the subject of Energy Harvesting Mechanism and Low Power Technology in Wireless Sensor Networks. More content can be added in the introduction part.

3.     Contributions in the introduction section should be highlighted in at least 4 bullet points to make it more readable.

4.     Authors can use latest related works from reputed journals from year 2022-23

5.     All the figures must be referenced in text in order. Figure 3(B) is referenced after the Figure 3(C) in text.

6.     Paper should be formatted as per the requirements of the journal.

7.     Conclusion should give complete overview of the work and study. It could be extended.

8.     Mention the future scope of your present works in the conclusion part.

9.     Read paper thoroughly for overall grammatical mistakes.

Author Response

Thank you for your comments. According with your advice, we amended the relevant part in manuscript. Some of your questions were answered below. Some of your questions were answered below.
1.Technically, the paper is well written.
Response: Thank you for your comments. Those kind comments are all valuable and very helpful for revising and improving our paper, as well as the important guiding significance to our researches. 

2.Researchers have put forward an effective strategy on the subject of Energy Harvesting Mechanism and Low Power Technology in Wireless Sensor Networks. More content can be added in the introduction part. Contributions in the introduction section should be highlighted in at least 4 bullet points to make it more readable.
Response: Thank you for your guidance. The introduction part has been added the content of research work. However, in terms of contribution, we really do not know what to add, and we can only start further research work in the future. 

3.Authors can use latest related works from reputed journals from year 2022-23.
Response: Thank you for your suggestions. We amended the relevant part in manuscript.

4. All the figures must be referenced in text in order. Figure 3(B) is referenced after the Figure 3(C) in text.
Response: We were sorry for our careless mistakes. This problem have been corrected. 

5.Paper should be formatted as per the requirements of the journal. Read paper thoroughly for overall grammatical mistakes.
Response: Thank you for pointing this out. We have carefully revised the manuscript.

6.Conclusion should give complete overview of the work and study. It could be extended. Mention the future scope of your present works in the conclusion part.
Response: Thank you for your suggestions. We amended the relevant part in manuscript.

Reviewer 3 Report

Comments and Suggestions for Authors

I appreciate the authors adressing some of my previous comments, however the applied revisions cannot be considered as sufficient. The added part in chapter 2 and 3 dealing with energy harvesting technologies and WSN is very general without any focus or connection to the presented contribution's topic of research, and relevant references to support statements in chapter 3 are missing. Revision of chapter 4 confronting piezoelectric cantilever resonator does not address my previous comment, as it does not contain information about excitation and generated power, only a generally known relation between cantilever size and resonant frequency, as well as voltage but without any information about current electric load. 

I do not believe that the presented revisions justify the contribution to be published and I would recommend the authors to narrow down their area of focus and extend their simulations / experiments to cover all related areas of lower power WSN.

Comments on the Quality of English Language

Compared to the initially submitted version of the contribution, English language has not been sufficiently improved.

Author Response

Thank you for your valuable and helpful comments on our paper. We greatly appreciate the guidance they provide for revising and improving our work. Based on your advice, we have made revisions to the Related Work, Simulations and Results, and Conclusion sections of the manuscript. Additionally, we have thoroughly reviewed and corrected any grammatical errors in the text. We apologize for any shortcomings in our research. Moving forward, we will continue to explore the bistable piezoelectric cantilever power generation system, develop a dynamic model for this system, further investigate its response characteristics under random excitation, and optimize its design.

Reviewer 4 Report

Comments and Suggestions for Authors

Thank you for addressing my comments and concerns carefully.

The revised version of the manuscript is ready for publication. However, before submission please make sure that the format of all references is consistent. Currently, it seems like there are two different reference formats included, but this could be due to the differences in the journals, I am not very sure of the reason. Nevertheless, to improve the quality of the manuscript, all references should reflect the same format, so please double-check this issue before resubmission.

Other than that, this manuscript is ready for publication. Great job!

The manuscript is ready for publics

Comments on the Quality of English Language

The level of English is improved and satisfactory.

Author Response

Thank you for your comments. Those kind comments are all valuable and very helpful for revising and improving our paper, as well as the important guiding significance to our researches. According with your advice, we have revised the references in revised manuscript. In addition, we have carefully examined and corrected the grammatical problems of the manuscript.